# New Evidences about the Carcinogenic Effects of Ochratoxin A and Possible Prevention by Target Feed Additives

**DOI:** 10.3390/toxins14060380

**Published:** 2022-05-30

**Authors:** Stoycho D. Stoev

**Affiliations:** Department of General and Clinical Pathology, Faculty of Veterinary Medicine, Trakia University, Students Campus, 6000 Stara Zagora, Bulgaria; s_stoev@hotmail.com

**Keywords:** ochratoxin A, carcinogenic effect, prevention, hygiene control, risk assessment, feed additives

## Abstract

A review of the carcinogenic effects of ochratoxin A (OTA) on various tissues and internal organs in laboratory and farm animals is made. Suggestions are made regarding how to recognize and differentiate the common spontaneous neoplastic changes characteristic for advanced age and the characteristic neoplasia in different tissues and organs in laboratory animals/poultry exposed to OTA. The synergistic effects of OTA together with its natural combination of penicillic acid are also investigated regarding possible carcinogenic effects. The malignancy and the target location of OTA-induced neoplasia is studied. The sex-differences of such neoplasia are investigated in the available literature. The time of appearance of the first neoplasia is investigated in long-term carcinogenic studies with OTA-treated animals. The possibility of target feed additives or herbs to counteract the toxic and carcinogenic effects of OTA is studied in the available literature. Some effective manners of prophylaxis and/or prevention against OTA contamination of feedstuffs/foods or animal production are suggested. The suitability of various laboratory animals to serve as experimental model for humans with regard to OTA-induced tumorigenesis is investigated.

## 1. Introduction

Most mycotoxins are well-known to possess various adverse effects, e.g., carcinogenic, cytotoxic and genotoxic effects on animals, poultry and humans [1]. Ochratoxin A (OTA), which is a frequent contaminant in feeds/foods for animals and humans all over the world [1,2,3], is defined by IARC as a possible human carcinogen and belongs to group 2B mycotoxins [4,5], which is based on various experimental studies with rats or mice [6,7,8,9]. Therefore, there are still some disputes among the research community in regard to the real potential of OTA to be carcinogenic for animals, poultry or humans because the mentioned above studies address mainly laboratory animals, such as mice or rats, whereas the data for humans, farm animals and poultry are still limited [10,11].

Some authors insist that OTA does not have carcinogenic effects and possesses only teratogenic action [12]; however, this opinion is disputed by other researchers [6,11,13,14,15,16,17,18,19,20]. Some recent 2-year studies in mice and rats clearly revealed the strong carcinogenic effect of OTA and classified different kind of induced tumors [21,22]. The carcinogenic effect of OTA has been reported by many research teams; however, its combined carcinogenicity with other mycotoxins in proportions similar to these in practice has been scarcely investigated [11,21]. 

There is scarce information in regard to the real carcinogenic effects of OTA, when it is ingested together with some other mycotoxins, e.g., penicillic acid (PA) (as this happens in practice) and whether there is a synergistic effect between the same mycotoxins in regard to their carcinogenicity. This is an important issue as it is well known that the bacterial microflora of the cecum in mice and rats is responsible for OTA hydrolyzation into the nontoxic metabolite OTα [23]. This hydrolyzation is performed mainly by the enzymes carboxypeptidase A and chymotrypsin [24]. However, some authors reported that PA can inhibit the activity of carboxypeptidase “in vitro” and “in vivo” [25], which could damage the mechanism of OTA detoxification and could contribute to its stronger toxicity [26,27] and carcinogenicity [21] when ingested together with PA as happens in practice.

Most of the available data only addresses the synergistic effects between OTA and PA [28,29,30,31] or fumonisin B1 (FB1) [32] in regard to their toxic but not carcinogenic effects. There is only one study that clearly shows the increased carcinogenic effects of OTA when it is administered together with PA as happens in practice [21]. This is a disturbing circumstance because mycological investigations of feeds or foods often reveal that PA [33,34] or PA-producing strains, such as *Penicillium* species, e.g., *P. aurantiogriseum,* often contaminate animal feeds or human foods [10,33,34]. 

Moreover, PA is usually produced by the same ochratoxinogenic fungi, e.g., *Aspergillus* strains, such as *Aspergillus ochraceus*, which are known to be the major producers of OTA in the warmer regions of the world [10,33,34,35]. This is why the multiple mycotoxin contamination in animal feeds or human foods is a real concern and could be a serious hazard for animals or humans around the world as the available information and knowledge about the carcinogenic effect of simultaneous mycotoxin exposure is extremely limited [21]. The synergistic effects between the same mycotoxins is well known but only in regard to their toxic effect on the kidneys, immune system and certain other organs [26,27,28,31] and not regarding their carcinogenic effects.

The potent OTA-toxicity in the same studies [26,27,28] has been found to result from unique synergistic interactions between OTA and PA and is reported in mice [36,37], poultry [27,29] or pigs [26]. The real cause for this synergism is reported to be the inhibition of pancreatic enzyme carboxypeptidase A by PA, which led to the subsequent impairment of OTA detoxification in the intestines [25]. The impairment of OTA detoxification is further responsible for the increased toxicity of OTA, when administered together with PA. 

Another disturbing circumstance is that PA can provoke functional damage in hepatobiliary excretion in mice [38], which in turn may also result in decreased hepatobiliary excretion of OTA and increased toxic or carcinogenic effects. Otherwise, the single PA administration has a low toxicity mainly on the liver [39], and only slight DNA breaks in mammalian cell lines [40] or inhibition of rat liver glutathione S-transferase activity [41] and disturbances in hepatobiliary excretion [38] have been reported so far. Therefore, the carcinogenic effect of OTA, when ingested with other target mycotoxins in proportions similar to these in practice, will be also reviewed and discussed in this review paper.

The possibility of target feed additives to counteract the toxic effects of OTA is often investigated by the research community in order to provide a safe utilization of OTA-contaminated feed [11,22,27,42,43,44,45,46,47]. In this regard, some suggestions were made that the toxicity of OTA could be a result of a structural similarity of this mycotoxin with phenylalanine (PHE), which lead to a partial suppression of protein synthesis as a consequence of a concurrent competition in regards to target t-RNA [48,49,50,51,52]. However, the possible protection of PHE has been investigated mostly in studies with target bacteria and yeast and only a few experiments are available in regards to some animals or poultry [11,22,43]. Therefore, the potential of PHE or some other feed additives to counteract carcinogenic effect of OTA will be reviewed and discussed in this review paper.

Some investigations will be also made in the available literature regarding whether laboratory animals could be an appropriate experimental model for humans. The malignancy and the target location of OTA-induced neoplasia will be also reviewed and discussed in regard to the male and female laboratory animals or poultry, in other to establish possible differences in the available literature. 

The possible ways to recognize and differentiate the common spontaneous neoplastic changes characteristic for advanced age and the typical neoplastic changes in different tissues and organs in laboratory animals/poultry exposed to OTA will be reviewed and discussed. Some effective manners of prophylaxis and/or prevention against OTA contamination of feeds/foods or animal production will be also reviewed and discussed.

## 2. Common Spontaneous Neoplastic Changes Characteristic for Advanced Age of the Respective Animals/Poultry

It is well-known that mice or rats are often used to evaluate the carcinogenic action of target toxic and carcinogenic substances. In this regard, historical data and strain sensitivities were considered to be important for adequate interpretation of the induced neoplasia [21,22]. The authors of such studies usually make a comparison between their results and the data from historical control groups in order to take into account the incidences of some spontaneously arisen tumors, characteristic for the advanced age of the used animal strain [53,54]. 

In this regard, it is important to evaluate only historical control data from the target animal species involved in the respective experiments and to use the same assessment criteria [54,55]. This approach is helpful in the case of evaluation of high incidence of uncommon neoplastic changes in the respective strain of mice/rats. For example, having in mind that above 90% of all spontaneous neoplasia in old Wistar rats are seen in the endocrine (42–48%), integumentary (26%), reproductive (9–11%) and hematopoietic/lymphatic (5.6–7%) systems [56,57,58], such neoplasia should not be considered to be specific as the same are not a consequence of carcinogenic effect of OTA; this mycotoxin is eliminated usually through the kidneys or liver and intestine, provoking direct toxic or neoplastic effects on the same tissues (Table 1). 

On the other hand, the neoplasia in the liver, kidneys, lung, skin and intestine should be attributed to the carcinogenic action of OTA as the same are not characteristic for the advanced age of rats and could be seen rarely (incidence is below 3–4%) [56,57,58]. Therefore, neoplasia in the urinary (below 0.4% in the rats of advanced age) or muscular (below 0.4% in the rats of advanced age) system, which are not characteristic for Wistar rats in advanced age [57,58,59], could be considered to be due to the cancerogenic effect of OTA [6,9]. 

It is known that squamous cell carcinoma of the eye is rarely seen in rats in advanced age [60]; however, the same tumor was recently found in rats exposed to 10 ppm OTA for a period of 2 years and is likely due to the carcinogenic effect of OTA or to a direct action of OTA-contaminated feed particles on the eyes’ surface [22]. The lung neoplasia could be also attributed to the intensive OTA-containing blood circulation in some organs, such as the lungs (Table 1).

The species and strain of the experimental mice are also important when investigate the carcinogenic effects of some substances. The BALB/c mouse is the most important laboratory strain for such experimental studies as neoplastic changes are uncommon and can be rarely found up to 22 months of age in BALB/c mice, mainly in the lung and mammary gland [61], and such tumors should be excluded as representative tumors in regards to the carcinogenicity of investigated substances, e.g., OTA. On the other hand, the mean lifespan of BALB/c mice sometimes can reach 29 months depending on the particular strain of mice [62], and thus the BALB/c strain is often used for long-term carcinogenic studies. 

In a recent study with BALB/c mice, the carcinogenic effect of OTA when combined with PA was also proven [21]. In regards to the common spontaneous tumors, only myoepithelioma, rhabdomyosarcoma, testicular tumors, lymphoma and mammary tumors were found with a low rate of 0.5–1% of the control BALB/c mice [63,64].

## 3. Characteristic and Specific Tumor Incidents and Neoplasia in Target Animals/Poultry Exposed to OTA

The pathological changes in early stages of experimental animals exposed to OTA usually were dominated by degenerative changes in the internal organs, which were in dependence of the level of OTA exposure and were well expressed in the kidneys, whereas the strong liver or intestinal damage and depletion of cells in lymphoid organs or fatty changes in the bone marrow were only characteristic for poultry [11,43,65] or mice and rats [21,22] exposed to OTA. 

In later stages (after 3 months’ OTA exposure or more), some proliferative changes were typically reported (e.g., connective tissue or mononuclear proliferation and activation of capillary endothelium) in the kidneys and sometimes in the liver, particularly well expressed in animals/chicks exposed to highest OTA-levels, and the same are thought to be a consequence of the strong degenerative changes in the same organs [11,22,27,43,44,45,46]. Neoplastic changes were also reported at the later stages of OTA exposure; however, the same were seen in some other tissues and organs, in addition to the target organs mentioned above (Figure 1, Figure 2 and Figure 3) [11,21,22].

Carcinogenic effects of OTA on the intestine, liver, kidneys, lung and eyes of Wistar rats were seen, when the same were exposed to feed levels of 5 or 10 ppm in their diet for a period of 2 years. The main degenerative changes in the same rats were reported to be stronger in the kidneys, liver, intestine, spleen and brain. In the same experiment, six of nine total neoplasms were reported to be malignant and three as benign [22].

The carcinogenic effects of OTA were also seen on muscles (Figure 1A), liver (Figure 1B), kidneys (Figure 1C), intestinal mesenterium (Figure 1D) and subcutaneous tissue (Figure 1E) of BALB/c albino mice exposed to 10 ppm OTA and/or 50–60 ppm PA via the feed and a total 22 neoplasias (14 were malignant and 8 benign) were seen during 20 months’ experimental period in three experimental groups (30 mice in each group) [21]. In the same experiment, the number of neoplasias was found to be significantly higher in the mice treated simultaneously with OTA and PA (14) as compared to mice exposed to OTA only (8), which revealed a strong synergistic effect between OTA and PA towards tumorigenesis. It is important to emphasize that the number of malignant neoplasias was reported to be higher (14) compared to benign ones (8) [21].

The target degenerative changes and neoplasms found in the liver, intestine, muscle, oculi and kidneys [11,21,22] were characteristic only for OTA-treated mice, rats or poultry as such tumors were not reported in pigs or humans exposed to OTA [10,15]. The same tumors and target damage are possibly a result of the route of OTA elimination and its distribution in the tissues of different animals (Table 1). In this regard, mice eliminate around 33% of ingested OTA via the hepatobiliary route of excretion [66] and the found neoplasia and damage in the liver and gastrointestinal tract of mice or rats could be attributed to the same enterohepatic recirculation of OTA in these animals and its repeated reabsorption via the gastrointestinal tract and further excretion through the liver [6,23,67,68]. The target tumors could be also due to the target degenerative changes in the same organs [69]. Unfortunately, the data and experimental studies in regards to the carcinogenic effects of OTA in poultry are scarce [11] and difficult to comment.

The recently reported tumors in the lung and eyes (Figure 2A) in OTA-exposed rats were explained by the inhalation of small OTA-contaminated feed particles through the dust or direct contact with the eye surface [15,22]. However, the same could be due to the OTA-containing blood supply for a long time because the plasma half-life of OTA in rats is long (55–120 h) [16] and because OTA possesses a long and constant carcinogenic and toxic action on the same tissues in rats through the blood circulation. 

The found neoplasia in the muscles (Figure 1A) and subcutaneous tissue (Figure 1E) in OTA-exposed mice could be a consequence of the same OTA circulation via the blood. The plasma half-life of OTA in mice is 24–39 h [67,68], and this likely also contributes to the continuous toxic and carcinogenic effect of OTA on the same tissues via the blood.

The first signs of carcinogenic effects of OTA are usually associated with the aneuploidy or polyploidy of nuclei and the polychromasia or polymorphism of the respective cells (Figure 2E,F), which could serve as an early indicator for subsequent carcinogenic transformation of the same cells and actually present their pre-carcinogenic state [17,70]. On the other hand, the frequent karyomegaly in the preserved cells adjacent to the observed neoplasia could serve as an indicator of the regenerative possibility of the same cells, which are usually found in the tissue around the damaged parenchyma [71,72] and could be involved in the elimination/detoxification of OTA.

The pericapillary edema and lymphatic enlargements around neoplastic areas is usually a consequence of OTA-induced vascular damage and increased permeability of vessels [11,21,22]. The observed benign adenomas in various tissues in OTA-treated animals could be considered as an initial stage of development of subsequent malignant tumors as reported by Kuiper-Goodman and Scot [73].

Some authors reported more frequent neoplasia in male rats or mice treated with OTA. [6,9,14,16,21,22,74,75]. For example, five of total six malignant neoplasms and two of total three benign neoplasms were found in male Wistar rats exposed to 5 or 10 ppm OTA for a period of 2 years [22]. 

Similarly, 9 of a total 14 malignant neoplasias and five of a total eight benign neoplasias were seen in the male BALB/c albino mice exposed to 10 ppm OTA and/or 50–60 ppm PA for a period of 20 months [21]. The authors explained such differences in the number of neoplasia in both sexes by sex-differences in target drag-metabolizing enzymes, which are involved in OTA-converting into some intermediate metabolites. Other authors explained these sex-differences with target hormonal differences between the both sexes [76].

There are various explanations about the carcinogenic potential of OTA, and some authors presented specific correlations between carcinogenic and genotoxic effects of OTA and particularly OTA-induced formation of DNA-adducts [77,78]. Another possible explanation about the carcinogenic effects of OTA could be associated with its immunosuppressive effect on both cellular and humoral immune response [31,79]. It is well-known that OTA exerts a strong suppression on the natural killer cells activity via the inhibition of endogenous interferon [80]. 

However, the same natural killer cells are involved in the regular destruction of tumor cells, and therefore the OTA-induced suppression of natural killer cell activity could further explain the actual mechanism for its carcinogenic effect. Some pathomorphological investigations revealed a strong reduction of the white pulp cells and degenerative changes in the lymph follicles of spleen in OTA-treated rats or mice [21,22], which could also explain its immunosuppressive effect on cellular immune response and subsequent carcinogenic effects. Such immunosuppression has also been reported in other studies with chicks or pigs treated with OTA, which subsequently developed target secondary bacterial infections [31,79].

In a recent experimental study, a larger number of malignant tumors was found using four times lower feed levels of OTA together with PA [21], compared with those described by other authors, who found such carcinogenic effects of OTA in mice using 40 ppm pure OTA [6,14,16], and surprisingly, benign tumors (adenomas) were found more often than malignant tumors. This circumstance suggests that some target mycotoxin combinations possess strong carcinogenicity at significantly lower mycotoxin levels.

Some other mycotoxins, such as fumonisin B1 [32], were also found to have a strong nephrotoxic effect on animal kidneys, which can add to the known nephrotoxic and carcinogenic effect of OTA. The same mycotoxins were also found in OTA-contaminated feeds in practice [33,34], which suggest the importance of the low levels of OTA that commercial chicks or pigs may encounter in some feeds.

Some recent experiments with rats and mice [21,22] clearly show that the same laboratory animals are not able to serve as experimental model for humans as suggested by some authors [17] because of different target organs of OTA-toxicity in humans and in rats/mice. In humans and pigs, the target organs of OTA toxicity are the kidneys [15,35], whereas significant damage and carcinogenic effects in rats, mice or poultry were seen in many other organs, e.g., the kidneys, liver, intestine, lung, subcutaneous tissue and muscles [11,21,22].

Of particular interest is the found carcinoma in the ureters of chick exposed to 5 ppm OTA for a period of 2 years (Figure 1F) [11], which is in good agreement with the reports about the increased frequency of carcinoma in the renal pelvis, ureters and urinary bladder in the patients suffering from Balkan Endemic Nephropathy (BEN) [81,82], which is supposed to be caused by OTA [15]. 

Such carcinomas have not been reported in laboratory and farm animals but, at the same time, are frequent findings in BEN-suffering humans [15] and were found to have DNA adducts similar to those seen in the kidneys of mice treated with OTA [83,84,85], which suggests a possible role of OTA in the development of these tumors of the urinary tract. The other tumors in chick exposed to 5 ppm OTA for a period of 2 years were found in kidneys (Figure 2D), liver (Figure 2C), spleen and muscles (Figure 2B) (Table 1) [11].

## 4. The Possible Preventive Measures of Some Feed Additives against Toxic and Carcinogenic Effects of OTA

The most potent protector against OTA toxicity was thought to be PHE as such a protective effect was found against an OTA-induced decrease in serum glucose or serum protein, in addition to the carcinogenic effect of OTA in rats. However, such protection was not always found with respect to OTA-induced changes in serum enzyme activity [22].

The histopathological changes in the Wistar rats exposed to 5 or 10 ppm OTA for a period of 2 weeks were seen in many internal organs; however, the same were not so strong in the rats additionally supplemented with PHE [22]. However, the protective effect of PHE against the carcinogenic effect of OTA was still controversial and only partially proven as seen from the number of OTA-induced neoplasia in the group of rats additionally supplemented with PHE. It is important to emphasize that the number of neoplasia in PHE-supplemented rats exposed to 10 ppm OTA via the feed is approximately the same as in the rats exposed to two times lower OTA concentration in the feed of 5 ppm, which suggests that protective effect of PHE is not as strong as expected [22].

Similarly, the reported carcinogenic or toxic effect of OTA on some internal organs and the suppressed egg production of Plymouth Rock hens was not ameliorated, when the same hens were supplemented with 25 ppm PHE in addition to 5 ppm OTA during the one-year experimental period [11,47]. Moreover, the number of OTA-induced neoplasias was similar between the group of hens additionally supplemented with PHE and the group only treated with the same feed level of 5 ppm OTA, which suggests that PHE cannot serve as a good protector against OTA carcinogenicity or against decreased egg production in hens [11,47].

It is thought that the strong toxicity of OTA is likely due to its structural homology with PHE and the subsequent suppression of protein synthesis or to the damaged production of tyrosine from PHE and also to the inhibition of enzymes participating in the PHE metabolism [51,86]. Some authors proposed that suppling the feed with enough PHE could prevent OTA-induced suppression of protein synthesis, which is responsible for the disruption of some lipoprotein formations, e.g., cellular membranes, lysosomes and mitochondria [22,51]. Such damage in membrane integrity could be responsible for the increased permeability of lysosomes and subsequent leakage of target autolytic enzymes into the cytosol of cells and the expected degenerative changes in the same cells. In this regard, feed supplementation with PHE should prevent or at least ameliorate the same cellular OTA provoked damage.

Unfortunately, some recent experiments with poultry and rats clearly showed that the protective effect of PHE was lower than the expectations of the authors [11,22,43,47]. A probable cause for such disruption of protective effect of PHE against OTA toxicity could be the circumstance that PHE-supplementation in mice was found to elevate the OTA-content in serum and liver by about four to eight times due to increasing its gastrointestinal absorption [67]. It appears that the observed decrease in protective effect of PHE against OTA toxicity could be due to the circumstance that the induced by PHE increase in the intestinal OTA-absorption counteract this protection. This is possibly the main cause for the slight protective effect of PHE as PHE cannot overwhelm the increased toxicity of the higher OTA levels.

In order to define the right protective measures against the toxic or carcinogenic effect of OTA, some other target mechanisms of OTA-toxicity have to be taken into account, e.g., the inhibition of mitochondrial transport and the increase in lipid peroxidation in addition to the observed genotoxic effect and DNA-adduct formation [16,73,78], which could be responsible for its carcinogenic effect. It was reported that OTA can suppress the macrophage phagocytic activity of natural killer cells and T-killer cells via decreasing of the basal inteferon and is partly responsible for cancerogenic effect of OTA [87]. 

The strong immunosuppressive effect of OTA on humoral and cellular immune response and the observed suppression of the activity in natural killer cell [31,79] could also disrupt the regular destroying of tumor cells [11,21,22]; however, the same should be prevented by target feed additives, e.g., immune boosters. The possible mechanism of immunosuppression could be due to the known inhibition of protein synthesis and subsequent delay of the division of the immunocompetent cells of the immune system provoked by OTA [87]. 

Such an impairment of protein synthesis in lymphocytes might lead to respective impairment in their proliferation and differentiation [88,89]. The OTA-induced suppression of humoral and cellular immune response, which is known in principle, was demonstrated in practice by an experimental study with pigs. It was shown that the immunosuppressive effect of OTA was the first expressed toxic effect of OTA, which induced the development of target secondary microbial infections in pigs at low contamination levels of 1 ppm OTA in the diet [79].

OTA was found to suppress cell-mediated and humoral immune responses in feed levels of 2–4 ppm in pigs [79,87,90] and chicks [91]; however, no immunosuppressive effect was found at feed levels of 1 ppm [43]. The immunosuppressive effect of OTA was seen to be stronger when chicks or animals were simultaneously treated with PA at low contamination levels of 0.2–0.8 ppm OTA [31], which suggested a stronger immunosuppressive and carcinogenic effect of the same toxins, when ingested together as happens in practice and was recently proven in mice experiments [21].

In such a way, the toxicity and carcinogenicity of various OTA-producing strains could be different and often not correlating with the capacity of OTA production. It seems that moldy feed contaminated with OTA and other mycotoxins as happen in the field could be much more dangerous because such feed was found to be more toxic or carcinogenic for laboratory and farm animals [21,26,31,32]. 

Having in mind that OTA is only a part of the complex of many other mold metabolites, e.g., PA, citrinine, and fumonisin B1, which might be synergistic with it in immunosuppressive or carcinogenic effects, special attention should be paid to the level of OTA interaction with other target metabolites in commercial chicken/pig diet or human food because such low contaminations levels of OTA could be dangerous when ingested together with other target metabolites in the same spontaneously contaminated feeds/foods [15,33,34,35].

In addition to PHE, significant protective effects against the growth inhibitory effects of OTA and associated pathomorphological changes were seen for some other feed additives, e.g., water extract of artichoke, sesame seeds (rich in PHE) and Roxazyme-G (polyenzyme complement produced by fungi ‘Trichoderma’) [27,31,43]. The protective effect of Roxazyme-G and sesame seeds was well expressed in regards to OTA-provoked changes in the kidneys and liver. 

Surprisingly, the protective effect of sesame seed and artichoke extract was found to be better expressed against 5 ppm OTA-induced suppression of humoral immune response as compared to PHE in the same experimental study. This could be due to the improved protein synthesis, which is damaged in OTA-treated animals and also to the improved division of the cells of the immune system, which is known to be destroyed in OTA-exposed animals as has been found [27,31,43]. 

That circumstance suggests that the same additives could be used to protect against the carcinogenic effects of OTA via protecting against the immunosuppressive effect of OTA on cellular immune response and the suppression of the activity in natural killer cells [31,79], which is responsible for the regular destruction of tumor cells [11,21,22]. The observed protection of the same additives against an OTA-induced decrease in the lymphoid organs’ weight in chicks suggests such a mechanism of protection [27,31,43].

Sesame seeds, which are rich in proteins (about 20%) and PHE (about 4.3%), present a cheaper way to supply animals with PHE and also to avoid the increased absorption of OTA from the gastrointestinal tract provoked by pure PHE [67]. Sesame seeds can also increase the energy metabolism in animals, which usually is damaged in OTA-exposed animals [92]. Similarly, Roxazyme-G can increase the digestive dissimilation of polysaccharides, which could improve digestible energy production by 8–13% and in such a way to improve the energy metabolism, which is impaired in OTA-treated animals.

On the other hand, artichoke-extract is recommended as a diuretic agent and was found to increase the urinary excretion of OTA [27,30,31,43]. It was also found that cynarine and flavonoids content in such extracts could increase the metabolism of cholesterol and decrease serum urea and lipids by improving the diuresis and biliary secretion, which can improve the hepatobiliary route of excretion of OTA in chicks (OTA is excreted via the bile and urine) [27,31,43]. In addition, cynarine and flavonoids content in artichoke-extract possess a strong protective effect on the liver [93,94,95] protecting against hepatocellular damages induced by OTA in chicks. The high levels of vitamin C in the artichoke-extract could also have a protective effect against OTA as it was found that ascorbic acid supplementation (300 mg/kg) to the diet of laying hens can reduce OTA toxicity [42].

Some experimental studies with poultry exposed to OTA and supplemented with various herbs or herbal products given as feed additives also revealed significant protection against the toxic and immunosuppressive effects of OTA. The OTA-induced damages in various internal organs and immunosuppression were less pronounced in the chicks/animals protected by certain target herbs or herbal products, e.g., *Silybum marianum, Withania somnifera* and Silymarin, whereas slight protection was seen for the herbs *Centella asiatica, Glycyrrhiza glabra* and *Tinospora cordifolia*. The protective effect of the same herbs was mainly seen on the kidneys, liver and immune system, which are usually compromised by OTA [44,45,46,96].

Clearly, some of the mentioned above protective additives, e.g., sesame seeds and artichoke-extract or target herbs or herbal products and extracts, could be used as possible supplements to the feeds in order to ameliorate the toxic and carcinogenic effects of OTA in animals or poultry. However, this possibility should be further investigated and analyzed, particularly in regards to the economic efficiency of each target additive to be used as a practical approach for safe utilization of OTA-contaminated fodder avoiding the condemnation of such fodder.

For preventing possible human exposure to this dangerous and relatively heat stable mycotoxin having a strong carcinogenic- and toxic effect on farm animals or humans, some preventive measures have been recently suggested to be undertaken at slaughterhouses [15,97,98]. For example, in order to prevent the contamination of chicken meat with OTA, the period of feed deprivation of chickens before slaughter could be prolonged. 

In addition, a possible change in the feed source for a several days in chicken or for a week in pigs before slaughter time in farms with mycotoxic nephropathy could decrease tissue contamination with OTA because OTA has a short half-life in chicks (4 h) [43] and pigs (72–120 h) [15]. In already slaughtered pigs and chicks, the prevention of tissue contamination with OTA may include a condemnation of the kidneys and liver (in chicks) or kidneys (in pigs) as OTA is accumulated in the same target organs in high levels.

## 5. Concluding Remarks

The available literature revealed a proven carcinogenic effect of OTA on the kidneys, liver, intestine and intestinal mesenterium as the same organs participate in the elimination or detoxification of OTA in different laboratory animals (e.g., hepatobiliary rout of excretion and enterohepatic recirculation of OTA in mice, rats and poultry). On the other hand, the neoplastic changes seen in endocrine, integumentary, reproductive and hematopoietic/lymphatic systems [56,57,58] should not be considered as specific because they are not a consequence of the carcinogenic effects of OTA but should be attributed to advanced age (Table 1). 

The observed neoplasia in the lung, ureters, subcutaneous tissue, muscle and eyes (in addition to the above-mentioned neoplasia) should be partially attributed to the carcinogenic effect of OTA as the same are not characteristic for the advanced age of rats/mice or poultry (Table 1). Such neoplasia could be due to a direct action of OTA-contaminated feed particles on the eyes’ surface (neoplasia in the oculi) or the inhalation of such OTA-contaminated feed particles through dust (lung neoplasia) and continuous OTA circulation via the blood (muscle neoplasia). On the other hand, degenerative changes in the same tissues/organs, which were provoked by OTA-exposure could also contribute to the initiation of such neoplasia.

The strong synergistic effect between OTA and PA, reported recently [21] in regard to tumorigenesis, was also an important issue because the both mycotoxins have been often found together in spontaneously contaminated feed/food. The more frequent neoplasia in male rats/mice exposed to OTA, as reported in the literature, could be explained by sex-differences in the target drag-metabolizing enzymes, which are involved in OTA conversion into certain intermediate metabolites [76]. The strong immunosuppressive effect on both the cellular and humoral immune response [31,79] could significantly contribute to the carcinogenic effect of OTA because natural killer cells are involved in the regular destruction of tumor cells.

Recent experiments with rats and mice [21,22] suggested that these laboratory animals are not appropriate to be an experimental model for humans because of the differences in the target organs of OTA-toxicity in humans (kidneys) and in rats/mice (kidneys, liver, intestine, lung, etc.). However, the observed carcinoma in the ureters in chicks exposed to OTA is in good agreement with the reports about the increased frequency of carcinoma in the renal pelvis, ureters and urinary bladder in patients suffering from Balkan Endemic Nephropathy [15].

The protective effect of PHE (known to be a structural analog of OTA) against the carcinogenic effect of OTA is still controversial and only partially proven via two recent experiments [11,22], which suggests that PHE cannot serve as a good protector against OTA carcinogenicity. A possible explanation could be that PHE-supplementation was found to elevate the serum OTA-level about four to eight times due to increasing its gastrointestinal absorption [67], which could counteract this protection. 

In this regard, some other feed additives found to protect against the toxic and immunosuppressive effects of OTA [31,43,44,45,46], e.g., Roxazyme-G, artichoke-extract, sesame seed (containing high levels of PHE), the herbs *Silybum marianum* and *Withania somnifera* and the herbal extract Silymarin, should be further investigated for possible protective effects against OTA-carcinogenicity.

## Figures and Tables

**Figure 1 toxins-14-00380-f001:**
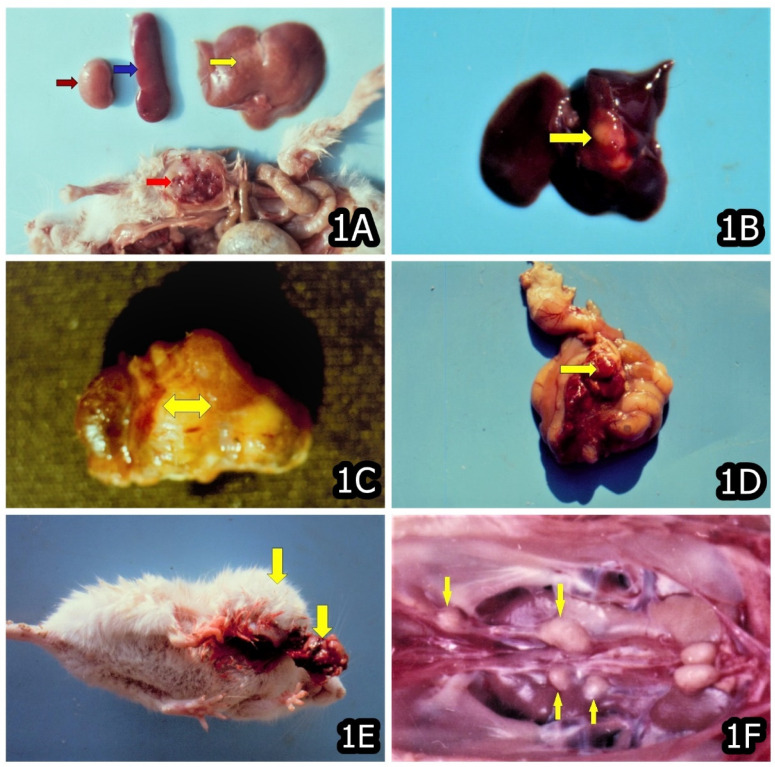
(**A**) Rhabdomyosarcoma in the muscles (red arrow), necroses in the liver (yellow arrow) and enlargement of the spleen (blue arrow) and pale color of kidney (brown arrow) in a mouse that died between months 15–20 and was exposed to 10 ppm OTA and 50–60 ppm PA. (**B**) Carcinoma in the liver (yellow arrow) in a mouse that died between months 10–15 and was exposed to 10 ppm OTA and 50–60 ppm PA. (**C**) Carcinoma in the region of kidney (yellow arrow) in a mouse that died between months 15–20 and was exposed to 10 ppm OTA and 50–60 ppm PA. (**D**) Angiosarcoma in the intestinal mesenterium (yellow arrow) in a mouse that died between months 15–20 and was exposed to 10 ppm OTA and 50–60 ppm PA. (**E**) Subcutaneous sarcoma (yellow arrows) in a mouse that died between months 15–20 and was exposed to 10 ppm OTA. (**F**) Carcinoma in the region of ureters (yellow arrows) of male chick exposed to 5 ppm OTA via the feed, which died at the end of the 20th month of the experiment. Large grey-white neoplastic foci are seen along the ureters and protruded significantly above its surface.

**Figure 2 toxins-14-00380-f002:**
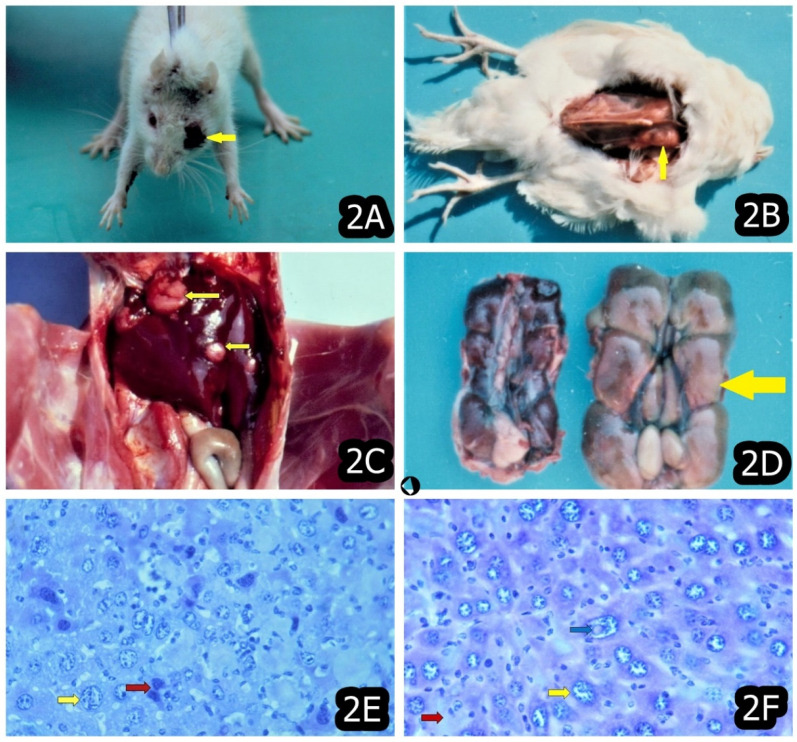
(**A**) Squamous cell carcinoma in the eye (yellow arrow) of a rat exposed to 10 ppm OTA via the feed, which was slaughtered at the end of the 24th month of the experiment. The neoplasia spread over the eyelid, conjunctiva and cornea. (**B**) Rabdomyoma in the breast muscle (yellow arrow) of female chick exposed to 5 ppm OTA and 25 ppm PHE via the feed, which was slaughtered at the end of the 24th month of the experiment. Large neoplasia in the region of breast muscle, which protruded significantly above the surface. (**C**) Adenocarcinoma in the liver (yellow arrows) of male chick exposed to 5 ppm OTA via the feed, which died at the end of the 10th month of the experiment. Large grey-white neoplastic foci in the diaphragmatic surface of the liver, which protruded significantly above the surface. (**D**) Lymphosarcoma in the kidneys (yellow arrow) of male chick exposed to 5 ppm OTA via the feed, which died at the end of the 18th month of the experiment (right). Kidneys without neoplastic changes in the chick from the same group (left). (**E**) Karyomegaly/aneuploidy (yellow arrow) and polychromasia (brown arrow) of nuclei of hepatocytes in the liver in a mouse at the end of the third month of exposure to 10 ppm OTA and 50–60 ppm PA. (**F**) Karyomegaly/aneuploidy (yellow arrow) of nuclei and polychromasia of hepatocytes (brown arrow). Nuclear acidophilic inclusions (blue arrow) in hepatocytes of the liver in a mouse at the end of the third month of exposure to 10 ppm OTA and 50–60 ppm PA.

**Figure 3 toxins-14-00380-f003:**
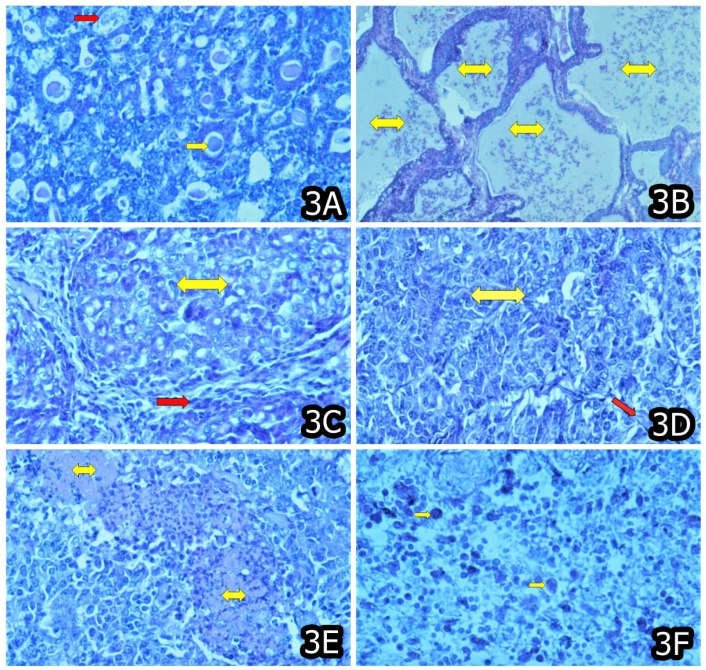
(**A**) Adenocarcinoma in the kidney containing secreted substance or hyaline (yellow arrow) and and/or a few necrotic tumor cells (red arrow) in a mouse at the end of 20 months exposure to 10 ppm OTA and 50–60 ppm PA. (**B**) Adenocarcinoma lined with several layers of neoplastic cells and containing a secreted substance and/or necrotic tumor cells and leucocytes (yellow arrow) in the liver in a mouse that died between months 15–20 and was exposed to 10 ppm OTA and 50–60 ppm PA. (**C**) Adenocarcinoma (nest adenocarcinoma—yellow arrow) in the liver with well-formed nests of malignant tumor cells surrounded by a relatively well-developed stromal tissue (red arrow) in a mouse that died between months 10–15 and was exposed to 10 ppm OTA and 50–60 ppm PA. (**D**) Carcinoma solidum (yellow arrow) with scarce stromal tissue (red arrow) in the liver of a mouse that died between months 10–15 and was exposed to 10 ppm OTA and 50–60 ppm PA. (**E**) Carcinoma solidum with many necroses (yellow arrow) among the neoplastic tissue infiltrated with leucocytes in the liver of a mouse that died between months 10–15 and was exposed to 10 ppm OTA and 50–60 ppm PA. (**F**) Sarcoma mixtocellulare consisted of globular cells with different size, polymorphism, polychromasia (yellow arrow), low differentiation, many irregular mitoses of the cells and their nuclei in the subcutaneous tissue in a mouse that died between months 15–20 and was exposed to 10 ppm OTA.

**Table 1 toxins-14-00380-t001:** Neoplasia in various tissues and organs in experimental laboratory animals or poultry and their attributability to OTA.

Neoplasia in Various Tissues and Organs in Experimental Laboratory Animals or Poultry		
Mainly attributed to OTA	Possibly attributed to OTA	Not attributed to OTA but to age
References [6,9,11,14,21,22]	References [11,21,22]	References [53,54,55,56,57,58]
kidneys	lung	endocrine system
liver	ureters	reproductive system
intestine	subcutaneous tissue	hematopoietic system
intestinal mesenterium	muscle	lymphatic system
	eyes	integumentary system

## Data Availability

Not applicable.

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
