# Peer review of "New Evidences about the Carcinogenic Effects of Ochratoxin A and Possible Prevention by Target Feed Additives"

_toxins, 2022, doi:10.3390/toxins14060380_

Round 1

Reviewer 1 Report

I have carefully revised the manuscript by ‘New evidences about the carcinogenic effect of ochratoxin A and a possible prevention by target feed additives’ and I believe that the work is of interest for the field and well planned and organized. The introduction reports sufficient information and the subject of each paragraph is well addressed. I have only few suggestions before possible publication on Toxins journal.

-Please check the correct verb form in the first sentence of the abstract and revise accordingly. Moreover, I suggest to revise the abstract because some parts are difficult to understand respect to the main text;

-The organization of both Figures and Figures legend is difficult to understand. I actually see three Figures with 6 panels for each Figure (not 18 Figures). Please revise all, using the letters for each panel (e. g. Fig. 1a, 1b and so on), reporting a Figure legend for each Figure on the bottom of them. Additionally, the authors could summarize the information also in Table form.

-I suggest to add a brief section for concluding remarks to better highlight the importance of this contribution;

-Please revise the references accordingly to the journal guidelines: e. g. year in bold; abbreviated journal name in Italics, volume in Italics and so on.

Author Response

Reviewer 1

I have carefully revised the manuscript by ‘New evidences about the carcinogenic effect of ochratoxin A and a possible prevention by target feed additives’ and I believe that the work is of interest for the field and well planned and organized. The introduction reports sufficient information and the subject of each paragraph is well addressed. I have only few suggestions before possible publication on Toxins journal.

A detailed Answer to the raised questions of Referee 1

All changes in the manuscript are now highlighted via using the option “track changes” or/and marked in yellow color.

  1. Please check the correct verb form in the first sentence of the abstract and revise accordingly. Moreover, I suggest to revise the abstract because some parts are difficult to understand respect to the main text;

-Answer: The correct verb form was now used in the first sentence of the abstract as referee noted. The abstract is now revised in order to facilitate the readers and to make it more comprehensible for them. Some of the sentences are simplified.

  1. The organization of both Figures and Figures legend is difficult to understand. I actually see three Figures with 6 panels for each Figure (not 18 Figures). Please revise all, using the letters for each panel (e. g. Fig. 1a, 1b and so on), reporting a Figure legend for each Figure on the bottom of them. Additionally, the authors could summarize the information also in Table form.

-Answer: The organization of both Figures and Figures legend is now changed according to Referee suggestions via using the letters for each panel (e.g. Fig. 1A, 1B, etc. The Figure legend for each Figure is now placed on the bottom of the Figure according to referee suggestions.

Also a new Table 1 is now prepared in order to summarize some of the important information as referee suggested.

  1. I suggest to add a brief section for concluding remarks to better highlight the importance of this contribution;

-Answer: A brief section Concluding remarks is now prepared in order to summarize each item in an easy-to-understand manner and to highlight the importance of this contribution as referee suggested – see below:

“The available literature revealed a proven carcinogenic effect of OTA on kidneys, liver, intestine or intestinal mesenterium, as the same organs participate in elimination or detoxification of OTA in different laboratory animals (e.g. hepatobiliary rout of excretion and enterohepatic recirculation of OTA in mice, rats or poultry). On the other hand, the neoplastic changes seen in endocrine, integumentary, reproductive and hematopoietic/lymphatic systems [56-58] should not be considered as specific, because the same are not a consequence of carcinogenic effect of OTA, but should be attributed to the advanced age. The observed neoplasia in the lung, ureters, subcutaneous tissue, muscle and eyes (in addition to the above mentioned neoplasia) should be partially attributed to the carcinogenic effect of OTA as the same are not characteristic for the advanced age of rats/mice or poultry. Such neoplasia could be due to a direct action of OTA-contaminated feed particles on the eyes’ surface (neoplasia in the oculi) or the inhalation of such OTA-contaminated feed particles through the dust (lung neoplasia) and continuous OTA circulation via the blood (muscle neoplasia). On the other hand, degenerative changes in the same tissues/organs, which were provoked by OTA-exposure could also contribute to the initiation of such neoplasia.

The strong synergistic effect between OTA and PA, reported recently [21] in regard to tumorigenesis was also an important issue, because the both mycotoxins have been often found together in spontaneously contaminated feed/food. The more frequent neoplasia in male rats/mice exposed to OTA, reported in the literature, could be explained by some sex-differences in target drag-metabolizing enzymes, which are involved in OTA-converting into some intermediate metabolites [76]. The strong immunosuppressive effect on both cellular and humoral immune response [31,79] could significantly contribute to carcinogenic effect of OTA, because the natural killer cells are involved in the regular destruction of tumor cells.

The recent experiments with rats and mice [21,22] suggested that the laboratory animals are not appropriate to be an experimental model for humans, because of the difference in the target organs of OTA-toxicity in humans (kidneys) and in rats/mice (kidneys, liver, intestine, lung, etc). However, the observed carcinoma in the ureters in chicks exposed to OTA is in good agreement with the reports about the increased frequency of carcinoma in the renal pelvis, ureters and urinary bladder in the patients suffering from Balkan Endemic Nephropathy [15].

The protective effect of PHE (known to be a structural analog of OTA) against the carcinogenic effect of OTA is still controversial and only partially proved via two recent experiments [11,22], which suggests that PHE cannot serve as a good protector against OTA carcinogenicity. A possible explanation could be that PHE-supplementation was found to elevate the serum OTA-level about 4-8 times, because of increasing its gastrointestinal absorption [67] and in such a way could counteract this protection. In this regard, some other feed additives found to protect against toxic and immunosuppressive effect of OTA [31,43-46], e.g. Roxazyme-G, Artichoke-extract, sesame seed (containing high levels of PHE), the herbs Silybum marianum, Withania somnifera and the herbal extract Silymarin, should be further investigated for possible protective effect against OTA-carcinogenicit.”

  1. Please revise the references accordingly to the journal guidelines: e. g. year in bold; abbreviated journal name in Italics, volume in Italics and so on.

-Answer: The reference is now revised accordingly to the journal guidelines: e. g. year in bold; abbreviated journal name in Italics, volume in Italics, names of the authors, etc. as referee suggested. 

Reviewer 2 Report

This manuscript has summarized the carcinogenic effects of ochratoxin A on various tissues and internal organs in laboratory or farm animals. This is an interesting topic. The following revision could improve the quality of the paper.

  1. The writing needs to be improved.
  2. Title, please correct “laboratory or farm animals” to “laboratory and farm animals”.
  3. Line 10, please correct “penicillic acid (PA)” to “”penicillic acid”. The abbreviation did not repeat in the abstract.
  4. Lines 28-30, please described the situation of occurrence/contamination of OTA in the food and feed by adding new references.
  5. Lines 77-79, it is better do not make only one sentence as one paragraph. Please combine them with the other paragraph.
  6. Line 147, “(Figures 1-18)”? It is not a right way to write the figues. All the other figures in the same figure need to be written as Figure 1A, B, C…etc. Please check the similar issues throughout the paper.
  7. Please move each figure to the corresponding main text. It is hard to the reviewer for the reading.
  8. Lines 193-195, it is better do not make only one sentence as one paragraph. Please combine them with the other paragraph. Please check throughout the paper.
  9. Lines 48, 155, 204, penicillic acid (PA) appeared in the different place. So, why you need the abbreviation?
  10. Please check the reference and make sure they are all right.

Author Response

Reviewer 2

This manuscript has summarized the carcinogenic effects of ochratoxin A on various tissues and internal organs in laboratory or farm animals. This is an interesting topic. The following revision could improve the quality of the paper.

A detailed Answer to the raised questions of Referee 2

All changes in the manuscript are now highlighted via using the option “track changes” or/and marked in yellow color.

  1. The writing needs to be improved.

-Answer: The writing is now improved as much as possible as referee suggested. Some of the sentences are simplified in order to facilitate the readers and to make the same sentences more comprehensible for readers, which have not got target competencies in some fields.

  1. Title, please correct “laboratory or farm animals” to “laboratory and farm animals”.

-Answer: The words “laboratory or farm animals” are now corrected to “laboratory and farm animals” as referee suggested throughout the entire manuscript.

  1. Line 10, please correct “penicillic acid (PA)” to “”penicillic acid”. The abbreviation did not repeat in the abstract.

-Answer: In line 10, the words “penicillic acid (PA)” is now corrected to ”penicillic acid”, because this abbreviation doesn’t not repeat in the abstract.

  1. Lines 28-30, please described the situation of occurrence/contamination of OTA in the food and feed by adding new references.

-Answer: Three more adequate references are now given in order to describe the situation of occurrence/contamination of OTA in the food and feed as referee suggested – see the same references below:

“Ochratoxin A (OTA), which is a frequent contaminant in feeds/foods for animals or hu-mans all over the world [1-3]…..”

  1. Streit, E.; Schatzmayr, G.; Tassis, P.; Tzika, E.; Marin, D.; Taranu, I.; Tabuc, C.; Nicolau, A.; Aprodu, I.; Puel, O.; Oswald, I. Current situation of mycotoxin contamination and co-occurrence in animal feed—focus on Europe. Toxins 2012, 4, 788-809.
  2. Pinotti, L.; Ottoboni, M.; Giromini, C.; Dell’Orto, V.; Cheli, F. Mycotoxin contamination in the EU feed supply chain: a focus on cereal byproducts, Toxins 2016, 8, 45, doi:10.3390/toxins8020045
  3. Schatzmayr, G.; Streit, E. Global occurrence of mycotoxins in the food and feed chain: facts and figures. World Mycotox. J. 2013, 6(3), 213-222

  1. Lines 77-79, it is better do not make only one sentence as one paragraph. Please combine them with the other paragraph.

-Answer: This paragraph is now combined with the previous paragraph as referee suggested.

  1. Line 147, “(Figures 1-18)”? It is not a right way to write the figures. All the other figures in the same figure need to be written as Figure 1A, B, C…etc. Please check the similar issues throughout the paper.

-Answer: The organization of both Figures and Figures legend is now changed according to Referee suggestions via using the letters for each panel (e.g. Fig. 1A, 1B, etc.)

  1. Please move each figure to the corresponding main text. It is hard to the reviewer for the reading.

-Answer: Each figure or combination of figures is now moved to the corresponding text as referee suggested, in order to facilitate the readers. 

  1. Lines 193-195, it is better do not make only one sentence as one paragraph. Please combine them with the other paragraph. Please check throughout the paper.

-Answer: This paragraph is now combined with the other paragraph as referee suggested. The same is did throughout the entire manuscript.

  1. Lines 48, 155, 204, penicillic acid (PA) appeared in the different place. So, why you need the abbreviation?

-Answer: The abbreviation of penicillic acid – PA, is now introduced throughout the entire manuscript after its firs explanation in line 48. The same is did with ochratoxin A and its abbreviation OTA as referee suggested. Thank you for seeing this inaccuracy.

  1. Please check the reference and make sure they are all right.

-Answer: The reference is now revised accordingly to the journal guidelines: e.g. year in bold; abbreviated journal name in Italics, volume in Italics, names of the authors, etc. as referee suggested. 

Reviewer 3 Report

This manuscript “New evidences about the carcinogenic effect of ochratoxin A and a possible prevention by target feed additives” reviews about ochratoxin A. Unfortunately, the text is hard to understand. In the introduction, it is necessary to summarize what will be discussed in this review and what will be discussed, item by item, in an easy-to-understand manner. Then, summarize each item in an easy-to-understand manner.

Author Response

Reviewer 3

A detailed Answer to the raised questions of Referee 3

All changes in the manuscript are now highlighted via using the option “track changes” or/and marked in yellow color.

  1. This manuscript “New evidences about the carcinogenic effect of ochratoxin A and a possible prevention by target feed additives” reviews about ochratoxin A. Unfortunately, the text is hard to understand.

-Answer: The writing is now improved as much as possible as referee suggested in order to facilitate the readers. Some of the sentences are simplified in order to make the same sentences more comprehensible for readers, which have not got target competencies in some fields.

  1. In the introduction, it is necessary to summarize what will be discussed in this review and what will be discussed, item by item, in an easy-to-understand manner. Then, summarize each item in an easy-to-understand manner.

-Answer: All items discussed in the review paper are now additionally summarized in the Introduction in a more comprehensive way as referee suggested – see the main changes below (the new text is marked by the “track changes” or in yellow color in the pape)r.

“…...... Therefore, the potential of PHE or some other feed additives to counteract carcinogenic ef-fect of OTA will be reviewed and discussed in this review paper.

Some investigations will be also made in the available literature, whether laboratory animals could be an appropriate experimental model for humans. The malignancy and the target location of OTA-induced neoplasia will be also reviewed and discussed in re-gard to the male and female laboratory animals or poultry, in other to establish possible differences in the available literature. The possible way to recognize and differentiate the common spontaneous neoplastic changes characteristic for advanced age and the typical neoplastic changes in different tissues and organs in laboratory animals/poultry exposed to OTA, will be reviewed and discussed. Some effective manners of prophylaxis and/or prevention against OTA contamination of feeds/foods or animal production will be also reviewed and discussed…..”

An additional attempt was made to highlight the main studied subjects in this review paper and to justify some further investigation in regard to carcinogenic effect of OTA ingested as a single mycotoxin or in combination with other target mycotoxins.

A brief section Concluding remarks is now prepared in order to summarize each item in an easy-to-understand manner and to highlight the importance of this contribution as referee suggested – see below:

“The available literature revealed a proven carcinogenic effect of OTA on kidneys, liver, intestine or intestinal mesenterium, as the same organs participate in elimination or detoxification of OTA in different laboratory animals (e.g. hepatobiliary rout of excretion and enterohepatic recirculation of OTA in mice, rats or poultry). On the other hand, the neoplastic changes seen in endocrine, integumentary, reproductive and hematopoietic/lymphatic systems [56-58] should not be considered as specific, because the same are not a consequence of carcinogenic effect of OTA, but should be attributed to the advanced age. The observed neoplasia in the lung, ureters, subcutaneous tissue, muscle and eyes (in addition to the above mentioned neoplasia) should be partially attributed to the carcinogenic effect of OTA as the same are not characteristic for the advanced age of rats/mice or poultry. Such neoplasia could be due to a direct action of OTA-contaminated feed particles on the eyes’ surface (neoplasia in the oculi) or the inhalation of such OTA-contaminated feed particles through the dust (lung neoplasia) and continuous OTA circulation via the blood (muscle neoplasia). On the other hand, degenerative changes in the same tissues/organs, which were provoked by OTA-exposure could also contribute to the initiation of such neoplasia.

The strong synergistic effect between OTA and PA, reported recently [21] in regard to tumorigenesis was also an important issue, because the both mycotoxins have been often found together in spontaneously contaminated feed/food. The more frequent neoplasia in male rats/mice exposed to OTA, reported in the literature, could be explained by some sex-differences in target drag-metabolizing enzymes, which are involved in OTA-converting into some intermediate metabolites [76]. The strong immunosuppressive effect on both cellular and humoral immune response [31,79] could significantly contribute to carcinogenic effect of OTA, because the natural killer cells are involved in the regular destruction of tumor cells.

The recent experiments with rats and mice [21,22] suggested that the laboratory animals are not appropriate to be an experimental model for humans, because of the difference in the target organs of OTA-toxicity in humans (kidneys) and in rats/mice (kidneys, liver, intestine, lung, etc). However, the observed carcinoma in the ureters in chicks exposed to OTA is in good agreement with the reports about the increased frequency of carcinoma in the renal pelvis, ureters and urinary bladder in the patients suffering from Balkan Endemic Nephropathy [15].

The protective effect of PHE (known to be a structural analog of OTA) against the carcinogenic effect of OTA is still controversial and only partially proved via two recent experiments [11,22], which suggests that PHE cannot serve as a good protector against OTA carcinogenicity. A possible explanation could be that PHE-supplementation was found to elevate the serum OTA-level about 4-8 times, because of increasing its gastrointestinal absorption [67] and in such a way could counteract this protection. In this regard, some other feed additives found to protect against toxic and immunosuppressive effect of OTA [31,43-46], e.g. Roxazyme-G, Artichoke-extract, sesame seed (containing high levels of PHE), the herbs Silybum marianum, Withania somnifera and the herbal extract Silymarin, should be further investigated for possible protective effect against OTA-carcinogenicit.”

Round 2

Reviewer 2 Report

No further comments. 

Reviewer 3 Report

The revised mansucript seems better.